# Mathematical Criteria for a Priori Performance Estimation of Activities of Daily Living Recognition

**DOI:** 10.3390/s22072439

**Published:** 2022-03-22

**Authors:** Florentin Delaine, Gregory Faraut

**Affiliations:** Université Paris-Saclay, ENS Paris-Saclay, LURPA, 91190 Gif-sur-Yvette, France; florentin.delaine@ens-paris-saclay.fr

**Keywords:** smart home, activity of daily living, activity modelling

## Abstract

Monitoring Activities of Daily Living (ADL) has become a major occupation to respond to the aging population and prevent frailty. To do this, the scientific community is using Machine Learning (ML) techniques to learn the lifestyle habits of people at home. The most-used formalism to represent the behaviour of the inhabitant is the Hidden Markov Model (HMM) or Probabilistic Finite Automata (PFA), where events streams are considered. A common decomposition to design ADL using a mathematical model is Activities–Actions–Events (AAE). In this paper, we propose mathematical criteria to evaluate a priori the performance of these instrumentations for the goals of ADL recognition. We also present a case study to illustrate the use of these criteria.

## 1. Introduction

In recent decades, life expectancy has increased in developed countries and may continue to increase in future. According to Reference [1], the percentage of the population aged 60 years and older will reach 30% in most of these countries by 2050. If these added years are lived in poor health, social-protection systems will face a major societal and economical challenge to take in dependent persons. Moreover, most people desire to age in place, creating a need for health-at-home systems to guarantee a sufficient level of security and a quality of life comparable to that whcih can be offered in care homes: this is Ambient Assisted Living (AAL).

To monitor health conditions at home, sensors must be deployed to acquire data such as physiological signals, environmental variables or the movements of the inhabitants. Various technologies can be used [2,3,4]: vision systems (cameras) [5,6,7,8], wearable devices (with on-board accelerometers, pedometers, heart rate sensors…) [9,10,11,12,13], binary sensors (motion detectors, door/window sensors, pressure sensors, etc.) [14,15,16,17,18,19], which can be combined into larger devices (smart floors with pressure sensors to detect falls) [20,21], or sensors that can be interpreted as n-ary with thresholds (water flow sensors, electricity consumption sensors, etc.) [22,23,24]. Some works have developed methods that consider sensor failures [25].

In this work, dedicated to human activity recognition (AR) [3,4], and especially the monitoring of activities of daily living (ADL), binary sensors and sensors with a signal that could be inferred as n-ary with thresholds are used. These technologies are usually inexpensive and minimally intrusive.

The recognition of an activity is high-level information that cannot be obtained directly from these sensors, which return semantically poor data. Therefore, activity models must be designed to build information from raw data.

To achieve this, Viard et al. [26,27,28], proposed using a semantic decomposition of activities, inspired by References [2,4]. This decomposition is either build by an expert or automatically built, for example, using pattern-mining techniques. However, the resulting decomposition may not be able to distinguish all the tracked activities. This could be due to movements that would allow for distinction but cannot be tracked by the deployed sensor network, e.g., more instruments would be required. Mistakes in the decomposition can also explain the difficulty of differentiating two activities due to errors made by the experts or the bias of inference algorithms.

Moreover, some works focus on metrics to evaluate the deviation of behavior [29,30,31]. Therefore, these works mainly focus on temporal criteria, and only focus on these for the deviation. This paper introduces metrics to evaluate the quality of the semantic decomposition of human activity at home. This aims to help build better activity descriptions. The metrics are meant to be used at an early stage, before any actual AR study.

In this paper, we propose a set of mathematical criteria to evaluate the effectiveness of AAL decomposition and instrumentation in a smart home. Many recognition works are instrumentation-dependant, and few of them compare instrumentation. Usually, the results are for the best AAL decomposition the authors found. Based on our previous work [26,27,28], which proposes a framework to discover and recognize activities, the set of criteria ensure a good effectiveness of the recognition framework.

Section 2 introduces the framework we use for the description of activities. Metrics are introduced in Section 3. A case study is presented in Section 5. Conclusions are drawn in Section 7, opening opportunities for future work.

## 2. Description of Activities

Activities are tasks performed by people. As in References [2,4], an activity can be considered as being composed of a set of actions. For instance, an activity such as “Taking care of his/her personal hygiene” can be related to actions such as “Having a shower”, “Brushing teeth”, etc. To be performed, actions require one or several movements. Movements can be observable or not, depending on whether a sensor can be deployed to detect the movement. The sensors considered in this work generate events. Thus, a set of events is associated with an action. This description is called the activities–actions–events (AAE) decomposition and is represented in Figure 1.

Note that, due to their low level of semantics, the information measured by sensors is considered as having the same level of semantics as elementary moves.

Moreover, as a movement can occur several times in an action, edges relating actions and events can be weighted in the graphical representation of the AAE decomposition. In this work, weighted edges are not considered, but their potential influence is discussed. In the same way, an action may belong to multiple activities. In this work, an action is considered as belonging to only one activity. The case in which this assumption is relaxed is discussed when it has an influence.

To monitor ADL, this decomposition must be designed. To carry this out, the use of expert knowledge or an unsupervised learning algorithm is proposed. In the case of an expert design, there are several possible approaches. If the activities to be monitored are known, it identifies the related actions and the movements composing them and then associates sensor events with the movements, either based on a given set of instruments or not. It is a top-down method. If the activities are not known, the goal is to identify activities based on a given set of sensor events, from which actions can be recognized, and then relate actions into activities. This approach is bottom-up. Unsupervised learning algorithms can be used to derive the AAE decomposition in a bottom-up manner.

Whatever the method chosen to design the AAE decomposition, there is currently no tool to determine if it will properly recognize the considered activities. As stated in Section 1, there may be mistakes or not enough tracked movements due to a lack of instruments. Thus, providing indicators and, ideally, a method of determining if an AAE decomposition is suitable for a given case of ADL monitoring would be extremely valuable. This is what we propose in the following section.

## 3. Metrics for the Analysis of Aae Decomposition

### 3.1. General Principle

In this section, the purpose is to define mathematical objects that can be automatically computed if an AAE decomposition is given. We provide five different kinds of metrics: the participation counters, the sharing rates, the weights, the elementary contributions and the distinguishabilities. Multiple metrics are proposed depending on the elements of the AAE decomposition at stake.

### 3.2. Notations

Let us define:$A_{i}$  is an activity. $A=\{A_{i}\}$ is the set of the considered activities$a_{j}$  is an action. $\sum_{a}\left(A_{i}\right)$ is the set of actions belonging to $A_{i}$ and $\sum_{a}\left(A\right)={⋃}_{A_{i}\in A}\sum_{a}\left(A_{i}\right)$$e_{k}$  is an event. $\sum_{e}\left(a_{j}\right)$ is the set of events belonging to $a_{j}$. It can also be defined for $A_{i}$ as $\sum_{a}\left(A_{i}\right)={⋃}_{a_{j}\in \sum_{a}\left(A_{i}\right)}\sum_{e}\left(a_{j}\right)$ and $\sum_{e}\left(A\right)={⋃}_{A_{i}\in A}\sum_{e}\left(A_{i}\right)$. Moreover, E is the set of all possible events and $\sum_{e}\left(A\right)\subseteq \sum_{E}$

### 3.3. Participation Counters

A first type of metric introduced is the participation counters. Their purpose is to count the actions and activities in which an event is involved. If actions can be involved in multiple activities, this can be also counted. These counters are defined mathematically as follows:

**Definition** **1.**
*The number of actions $a_{j}\in \sum_{a}\left(A_{i}\right)$ where $e_{k}\in \sum_{e}\left(a_{j}\right)$ is:*

$$C_{a}\left(e_{k}|A_{i}\right)=\sum_{a_{j}\in \sum_{a}\left(A_{i}\right)}\left|\left\{e_{k}\right\}\cap \sum_{e}\left(a_{j}\right)\right|$$



**Definition** **2.**
*The number of actions $a_{j}\in \sum_{a}\left(A\right)$ where $e_{k}\in \sum_{e}\left(a_{j}\right)$ is:*

$$C_{a}\left(e_{k}\right)=\sum_{a_{j}\in \sum_{a}\left(A\right)}\left|\left\{e_{k}\right\}\cap \sum_{e}\left(a_{j}\right)\right|$$



**Remark** **1.**
*For all $e_{k}\in \sum_{e}\left(A\right)$, $\sum_{A_{i}\in A}C_{a}\left(e_{k}|A_{i}\right)\leq C_{a}\left(e_{k}\right)$ Indeed, $\sum_{A_{i}\in A}|\sum_{a}\left(A_{i}\right)\left|\geq \right|\sum_{a}\left(A\right)|$.*


**Definition** **3.**
*The number of activities $A_{i}\in A$ where $e_{k}\in \sum_{e}\left(A_{i}\right)$ is:*

$$C_{A}\left(e_{k}\right)=\sum_{A_{i}\in A}\left|\left\{e_{k}\right\}\cap \sum_{e}\left(A_{i}\right)\right|$$



**Definition** **4.**
*The number of activities $A_{i}\in A$ where $a_{j}\in \sum_{a}\left(A_{i}\right)$ is:*

$$C_{A}\left(a_{j}\right)=\sum_{A_{i}\in A}\left|\left\{a_{j}\right\}\cap \sum_{a}\left(A_{i}\right)\right|$$



They allow to identify how many times all events or actions are involved in the AAE decomposition at any level. If an event is involved in many activities, its occurrence may not be significant in determining which activity was performed. On the contrary, if it involved in only one activity, it can be identified as a differenciating event using this metric.

Multiple events from the same activity can be involved in multiple other activities. However, their association can still allow activities to be properly recognized. Thus, these metrics are limited because they do not relate how activities may share actions and events, and how actions share events.

### 3.4. Sharing Rates

In order to identify how events and actions are shared in the AAE decomposition, sharing rates are defined. They offer a low-level reading of the similarities between activities or actions, e.g., what proportion of events are shared between different activities or actions. Once again, if actions can be related to more than one activity, a sharing rate between actions and activities can be defined. These rates are defined mathematically as follows:

**Definition** **5.**
*The rate of shared events of $a_{j}\in \sum_{a}\left(A_{i}\right)$ with $a_{k}\in \sum_{a}\left(A_{i}\right)$ is:*

$$S_{e}\left(a_{j}|a_{k}\right)=\frac{|\sum_{e}\left(a_{j}\right)\cap \sum_{e}\left(a_{k}\right)|}{|\sum_{e}\left(a_{j}\right)|}$$



**Definition** **6.**
*The rate of shared actions of $A_{i}\in A$ with $A_{j}\in A$ is:*

$$S_{a}\left(A_{i}|A_{j}\right)=\frac{|\sum_{a}\left(A_{i}\right)\cap \sum_{a}\left(A_{j}\right)|}{|\sum_{a}\left(A_{i}\right)|}$$



**Definition** **7.**
*The rate of shared events of $A_{i}\in A$ with $A_{j}\in A$ is:*

$$S_{e}\left(A_{i}|A_{j}\right)=\frac{|\sum_{e}\left(A_{i}\right)\cap \sum_{e}\left(A_{j}\right)|}{|\sum_{e}\left(A_{i}\right)|}$$



**Remark** **2.**
*A sharing rate can be applied versus multiple objects like:*

$$S_{e}\left(A_{i}|A_{j},A_{k}\right)=\frac{|\sum_{e}\left(A_{i}\right)\cap \left(\sum_{e}\left(A_{j}\right)\cup \sum_{e}\left(A_{k}\right)\right)|}{|\sum_{e}\left(A_{i}\right)|}$$



With sharing rates, similar activities or actions can be identified. Ideally, a sharing rate between two activities should be equal to zero, but this may rarely be possible in practice when monitoring multiple activities in the same room. With this type of metric, similar activities or actions can be identified, helping to detect possible issues at the recognition step. An activity that shares all its events with another activity may not be recognizable, depending on the chosen algorithm (both activities will be recognized with the algorithm from Reference [27]). When sharing rates are not equal to zero or one, it may be harder to conclude the possible issues related to the level of shared items. For instance, the sets of events associated with activities or actions can be of very different sizes, depending on their complexity. Thus, the intersection between sets of events of different activities can be more or less important in terms of size, but also in terms of consequences for recognition depending on the importance of the events for an activity. In this way, an acceptable sharing level must be defined for each activity. This can be different for each activity. Setting this level is not an easy task; this is discussed in Section 4.

Sharing rates are similar to participation counters in the idea. They differ in the perspective chosen for analysis. The first type considers the perspective of two or more items of a decomposition level regarding the elements of a lower level associated with them. The second one considers the perspective of an item of a level regarding elements of higher decomposition levels. Both these types of metric do not demonstrate how important an event is for an activity, e.g., offer an idea of the importance of the relationship between two items from different decomposition levels.

### 3.5. Weights

To quantify the role of an event in a particular activity or action, or the role of an action in a particular activity, weights are defined. They allow to identify which level an event or an action contributea to an activity regarding the size of event and action sets. They consist of associating a coefficient with an item that is inversely proportional to the size of its set.

If events/actions appear only once in each action/activity in which they are involved, weights are defined mathematically as follows:

**Definition** **8.**
*The weight of an event $e_{k}\in \sum_{e}\left(a_{j}\right)$ in $a_{j}\in \sum_{a}\left(A_{i}\right)$ is:*

$$W\left(e_{k}|a_{j}\right)=\frac{1}{|\sum_{e}\left(a_{j}\right)|}$$



**Definition** **9.**
*The weight of an action $a_{j}\in \sum_{a}\left(A_{i}\right)$ in $A_{i}\in A$ is:*

$$W\left(a_{j}|A_{i}\right)=\frac{1}{|\sum_{a}\left(A_{i}\right)|}$$



**Definition** **10.**
*The weight of an event $e_{k}\in \sum_{e}\left(A_{i}\right)$ in $A_{i}\in A$ is:*

$$W\left(e_{k}|A_{i}\right)=\sum_{a_{j}\in \sum_{a}\left(A_{i}\right)}W\left(e_{k}|a_{j}\right)W\left(a_{j}|A_{i}\right)$$



**Remark** **3.**
*If events or actions can appear multiple times in actions or activities, respectively (weighted edges in the AAE decomposition), the numerator is equal to the number of times it appears and the denominator is equal to the sum of all the events or actions.*


An event weight regarding an action or an activity can be more or less important depending on the size the associated event set. Looking at participation counters and weights for events enables us to determine if an event should be shared between activities. For instance, an event can have a high weight regarding an activity, e.g., few events contributes to the decomposition of the activity, and a high participation counter regarding activities, e.g., it is involved in many activities. Sharing rates should be studied to determine if the other events of the same activity are involved in multiple activities. It can also encourage more precise descriptions of an action, so that event weights decrease. Multiple conclusions are possible, as weights and all other metrics do not relate if an activity or action can be recognized among all the others.

### 3.6. Elementary Contributions

To define a metric allowing for an estimation of whether activity can be recognized, considering the decomposition of other activities, elementary contributions are defined. These are computed based on the participation counters. As for the weight of an event or an action regarding an activity, the idea is to estimate how important this is for a given activity, considering that it may be involved in other ones.

Elementary contributions are defined mathematically as follows:

**Definition** **11.**
*For $A_{i}\in A$, the elementary contribution of the event $e_{k}\in \sum_{e}\left(A_{i}\right)$ is:*

$$F\left(e_{k}|A_{i}\right)=\frac{C_{a}\left(e_{k}|A_{i}\right)}{\sum_{A_{i}\in A}C_{a}\left(e_{k}|A_{i}\right)}$$



**Definition** **12.**
*For $A_{i}\in A$, the elementary contribution of the action $a_{j}\in \sum_{a}\left(A_{i}\right)$ is:*

$$F\left(a_{j}|A_{i}\right)=\frac{1}{C_{A}\left(a_{j}\right)}$$



Particularly for $F(e_{k}|A_{i})$, this metric provides a more precise indication of the an event’s participation in an activity. It considers that the event might be involved in several actions. Regarding $F(a_{j}|A_{i})$, and as for $C_{A}\left(a_{j}\right)$, it is interested in whether actions are related to multiple activities.

### 3.7. Distinguishability

Considering elementary contributions, a natural idea is to sum all the contributions of events or actions related to an activity. This is what distinguishability is about. It aims to estimate a priori how well an activity can be differentiated, considering the perspective of actions or events. It is defined mathematically as:

**Definition** **13.**
*The distinguishability of an activity $A_{i}\in A$ in terms of events is:*

$$D_{e}\left(A_{i}\right)=\frac{1}{|\sum_{e}\left(A_{i}\right)|}\sum_{e_{k}\in \sum_{e}\left(A_{i}\right)}F\left(e_{k}|A_{i}\right)$$



**Definition** **14.**
*The distinguishability of an activity $A_{i}\in A$ in terms of actions is:*

$$D_{a}\left(A_{i}\right)=\frac{1}{|\sum_{a}\left(A_{i}\right)|}\sum_{a_{j}\in \sum_{a}\left(A_{i}\right)}F\left(a_{j}|A_{i}\right)$$



As a normalized sum of elementary contributions, distinguishabilities provide general metrics to demonstrate the extent to which one activity is related to all the others in a more balanced way than $S_{e}\left(A_{i}|\left\{A_{j}\right\}/A_{j}\in A\backslash \left\{A_{i}\right\}\right)$. It considers that there an event might be involved in several actions of the same activity.

An interesting theorem can be associated to these metrics:

**Theorem** **1.**
*$\forall A_{i}\in A$, if $D_{e}\left(A_{i}\right)=1$, then $D_{a}\left(A_{i}\right)=1$.*


**Proof.** Suppose $A_{i}\in A$ and $D_{e}\left(A_{i}\right)=1$.Then $\forall e_{k}\in \sum_{e}\left(A_{i}\right)$, $F(e_{k}|A_{i})=1$.Then $\sum_{e}\left(A_{i}\right)\cap \sum_{e}\left(A\backslash \left\{A_{i}\right\}\right)=Ø$Because $\sum_{e}\left(A_{i}\right)={⋃}_{a_{j}\in \sum_{a}\left(A_{i}\right)}\sum_{e}\left(a_{j}\right)$,Then $\sum_{a}\left(A_{i}\right)\cap \sum_{a}\left(A\backslash \left\{A_{i}\right\}\right)=Ø$.Then $\forall a_{j}\in \sum_{a}\left(A_{i}\right)$, $C_{A}\left(a_{j}\right)=1\Rightarrow F\left(a_{j}|A_{i}\right)=1$.Then $D_{a}\left(A_{i}\right)=1$ □

Thus, an activity that is identified as distinguishable in terms of event is also distinguishable in terms of actions.

**Remark** **4.**
*The converse is not true.*

*Suppose A and $A^{\prime }$ and $\sum_{a}\left(A\right)=a$, $\sum_{a}\left(A^{\prime }\right)=a^{\prime }$ and $\sum_{e}\left(a\right)\cup \sum_{e}\left(a^{\prime }\right)\neq Ø$.*

*Then, $D_{a}\left(A\right)=1$, but $D_{e}\left(A\right)<1$ because $\exists e\in \sum_{e}\left(A\right),F\left(e|A\right)<1$.*


Distinguishability is a powerful tool, completing the panel of presented metrics. Note that an activity that shares its events with another activity will not have a distinguishability equal to zero, but a low distinguishability. Moreover, as for sharing rates, activities can be judged as distinguishable from the others only if they reach an acceptable level. The same issues appear here: should it sharing rates be the same for each activity? How should they be set?

### 3.8. Conclusions

In this section, five different types of metrics are introduced. They look at an AAE decomposition from different points of view, e.g., activities, actions, and events, targeting a specific level of decomposition and with different considerations. Although it was shown by construction that each one is relevant to the study of AAE decomposition, it is not clear how they should be used. Thus, a methodology is needed to help such analyses.

## 4. Methodology for the Analysis of Aae Decomposition

### 4.1. General Principle

As the recognition of activities is targeted, a top-down approach is natural to analyze AAE decomposition. In this way, metrics considering activities in the most general manner as possible are studied first, and then, if issues are identified, other metrics related to actions and to events can be studied. Table 1 summarizes each metric and indicates the level of the AAE decomposition for which it contains information.

This study can be carried out during the design of AAE decomposition or after the identification of issues during Activity Recognition.

### 4.2. Detailed Methodology

The first metric we recommend studying is the distinguishability of each activity, in terms of events and actions. Sharing rates with the perspective of activities should also be considered. This will provide general information on the differences between the activities.

If the values of disintiguishabilities are judged to be too low for some activities, with a high eventual level of sharing rates in terms of actions or events, then the associated elementary contributions can be studied to determine which events or actions have the lowest values, and thus contribute the least to the distinguishability.

At this point, problematic events and actions are identified. With the participation of these events or actions, and their associated weights, the nature of the issue can be identified and its level of severity can be determined. This can be, for instance, an event with a high participation counter in activities, which is present in only one action of an activity that is not distinguishable. If it is possible to lower this activity’s weight by adding other events, then this may be a solution.

### 4.3. Acceptable Levels for Metrics

In our methodology, we do not provide threshold values for the metrics that help to make decisions such as declaring an activity as distinguishable. Indeed, this is a difficult task, depending on the number of activities, actions and events, and how they are distributed in the decomposition. We do not provide any recommendation at present, and proceed empirically and qualitatively more than quantitatively. We recall that our metrics are designed first to help an expert to design the decomposition, and help to check the results of an automated definition.

## 5. Case Study

This section introduces a case study based on experiments conducted in a real flat.

### 5.1. Smart Flat Description

To conduct the experiments, a two-room flat was provided by the ENS Paris-Saclay, France. Its photo is given in Figure 2, and a blueprint is represented in Figure 3. The flat can be decomposed into four zones to represent different spaces: the entrance, the bathroom, the kitchen and the bedroom (Figure 3).

### 5.2. Activities to Monitor and Proposed Decomposition

In this case study, three activities were monitored: cooking ($A_{C}$), the preparation of a hot beverage ($A_{HBP}$), and taking care of his/her personal hygiene ($A_{PH}$).

To represent their realization, two actions were defined per activity: “make tea” and “make coffee” for $A_{HBP}$, “prepare a ready-cooked dish” and “make pasta” for $A_{C}$, and “have a shower” and “go to the toilets” for $A_{PH}$. The number of actions was limited by the number of sensors in our possession at the time of the experiment: 8 door sensors, 3 door sensors with motion detectors, 5 smart outlets and 4 water-flow sensors.

Each instrument generates two events, corresponding to a rising edge and a falling edge of the binary signal, except for the PIR in the door sensors that detect motion, which return only rising edges. Therefore, 43 events were available to describe much movement as possible. Thirty-nine were used in the resulting decomposition represented in Figure 4. The sensors were deployed in the smart flat, as shown in Figure 3.

### 5.3. A Priori Estimation of Recognition

With the given AAE decomposition of Figure 4, the metrics introduced in Section 3 can be computed. For the sake of brevity, only activity -elated metrics, e.g., $S_{a}\left(A_{i}|A_{j}\right)$, $S_{e}\left(A_{i}|A_{j}\right)$, $D_{a}\left(A_{i}\right)$ and $D_{e}\left(A_{i}\right)$ for $A_{i},A_{j}\in A$, and event-related metrics for activities, e.g., $C_{a}\left(e_{k}|A_{i}\right)$, $C_{A}\left(e_{k}\right)$, $W(e_{k}|A_{i})$, $F(e_{k}|A_{i})$ for $e_{k}\in \sum_{e}\left(A\right)$ and $A_{i}\in A$, are considered. Their values are reported in Table 2 and Table 3, respectively.

The following observations can be made according to these results. According to Table 2, as no activities share any action, their actions are distinguishable. $A_{PH}$ is even distinguishable in terms of events, as it shares no events with the other activities. However, this is not the case for $A_{C}$ and $A_{HBP}$, which also have the same rate of shared events (0.56). However, $D_{e}\left(A_{HBP}\right)$ is slightly higher than $D_{e}\left(A_{C}\right)$, indicating that at least some events are shared in both $A_{HBP}$ actions but not both $A_{C}$ actions. Thus, these events may induce false positives when $A_{C}$ or $A_{HBP}$ are performed. Nevertheless, differentiating between the two activities should still be possible when considering the full sequence of events in an activity realization, as distinguishability is quite high for both.

By looking at Table 3, this observation is confirmed by shared events, such as “Kitchen|Cupboard_Bottom|Open|1”, which appears once in $A_{C}$ and twice in $A_{HBP}$ (according to $C_{a}\left(e_{k}|A_{i}\right)$ and $F(e_{k}|A_{i})$) and does not represent a major weight in the activity (0.04 for $A_{C}$, 0.08 for $A_{HBP}$).

In this way, the metrics proposed for a priori analysis allow for observations that could be major if confirmed by AR.

### 5.4. Experimental Dataset

To verify if our a priori observations were actually verified in practice, experimental data were acquired.

In the flat, the three activities were performed twenty times by the same user with some changes in the manner in which they were performed. A change can be an inversion of two steps, such as taking a tea bag before a cup when making tea. Noise was added by wandering into the room or interrupting an activity to perform another, as in a real situation. Interruptions were also introduced. Based on these realizations, a learning database was computed by selecting the desired number of activity instances, ordering them and concatenating them with the addition of random events between them $\sum_{e}\left(A\right)$. Finally, our training database was composed of 1737 events.

Test cases were also designed: simple test cases where only one action was performed in an usual way and complex test cases where multiple actions were realized with interruptions or parallel realizations.

### 5.5. Recognition Results

In this paper, $A_{i}$ is the PFA associated with activity $A_{i}$. Its set of events is $\sum_{A_{i}}=\sum_{e}\left(A_{i}\right)$. Moreover, states of the model represent actions; hence, their set $Q_{A_{i}}=\sum_{a}\left(A_{i}\right)$. PFA models were built and trained following the method presented in [26]. The AR was performed with the method presented in [27]. For both model learning and AR, sequences of five events at most were considered. This level was defined regarding the computational effort needed for AR. Activity estimation level was expressed by the inverse of the normalized likelihood.

Six test cases were considered—two per activity. AR results for each activity model are represented in Figure 5 with the declaration of realization for each activity.

First of all, according to Figure 5c,f, $A_{PH}$ was perfectly recognized with its associated model. The activity estimatior was greater than zero, whereas it was equal to zero with the other models. This confirms our a priori observation on $A_{PH}$ distinguishability.

Then, Figure 5a,b,d,e confirm that, due to their shared events, models $A_{C}$ and $A_{HBP}$ may simultaneously indicate that they are being realized. This confirms the observations made with event-sharing rates and distinguishability.

Identifying which activity is performed at first sight is not easy, except for the “Prepare a ready-cooked dish” test case (Figure 5a. Indeed, this action only has events with $A_{C}$ actions, which metrics like $S_{e}\left(a_{j}|a_{k}\right)$ would notice. Moreover, events in common in the actions “Make pasta”, “Make coffee” and “Make tea” provide the test cases in which false positives appear. False positives are more frequent than true positives in Figure 5a, as the events are seen in the actions of $A_{HBP}$, which is not the case for $A_{C}$. This validates the observations made for the common events. Finally, as parts of the AR estimation of $A_{HBP}$ are equal to zero, when this is not the case for $A_{C}$ in Figure 5a (during t=[161;191]), and parts of the AR estimation of $A_{C}$ are equal to zero when this is not the case for $A_{HBP}$ in Figure 5b,e (during t=[35;70] and t=[49;69] respectively), it is possible, with further analysis, to correctly identify the activity, confirming our a priori observation regarding the high levels of distinguishability for $A_{C}$ and $A_{HBP}$.

## 6. Discussion

Through the experimental case study, all the observations made a priori with the AAE decomposition and the presented metrics were confirmed during the AR step with our method. In this way, the introduced metrics appear relevant to help the study of decomposition, whether designed by an expert or obtained automatically. The proposed methodology for the analysis of the AAE decomposition provides help in managing the multiple metrics we introduced and can lead an user to appropriate conclusions.

Nevertheless, it is not possible to provide typical metric values for an AAE decomposition that would not raise any issues at AR. This is a major limitation to our approach, which is currently empirically driven.

## 7. Conclusions

In this paper, metrics for an a priori estimation of the performance of model-based Activity Recognition were introduced: participation counter, sharing rate, weight, elementary contribution and distinguishability. At all levels of the activity–action–event decomposition, they allowed to precisely determine how the resulting models will behave, along with their strengths and weaknesses. Metrics were computed on a real case study of Activity Recognition, in which the metrics were shown to be relevant in the analysis of AAE decomposition before deployment. Future work will be devoted to scaling up our approach, in order to automate it. Our current work focuses on a framework that will provide the best AAL decomposition by resolving the optimization problem. 

## Figures and Tables

**Figure 1 sensors-22-02439-f001:**
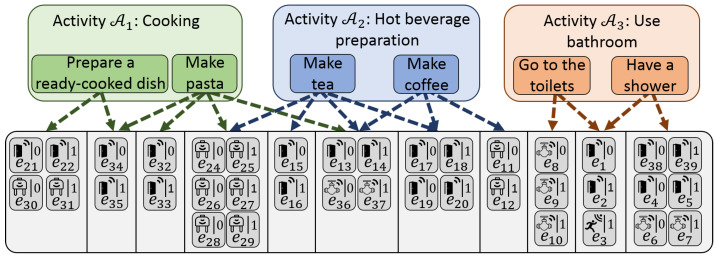
Activities-actions-events (AAE) decomposition of activities.

**Figure 2 sensors-22-02439-f002:**
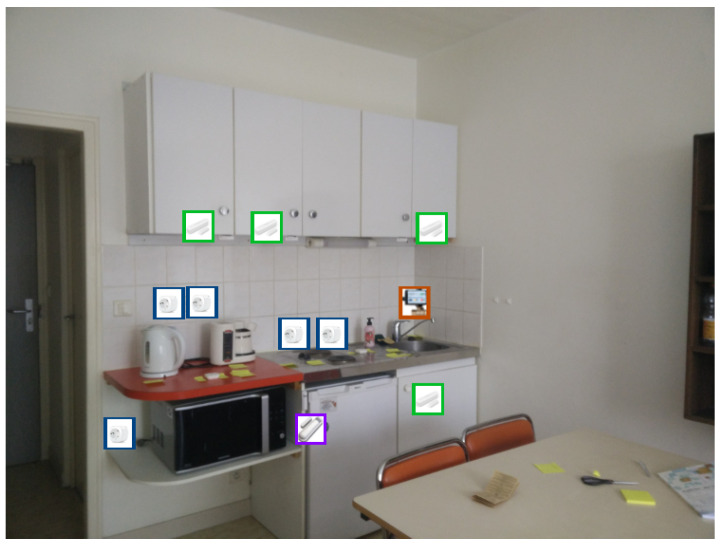
View of the kitchen with the position of the sensors.

**Figure 3 sensors-22-02439-f003:**
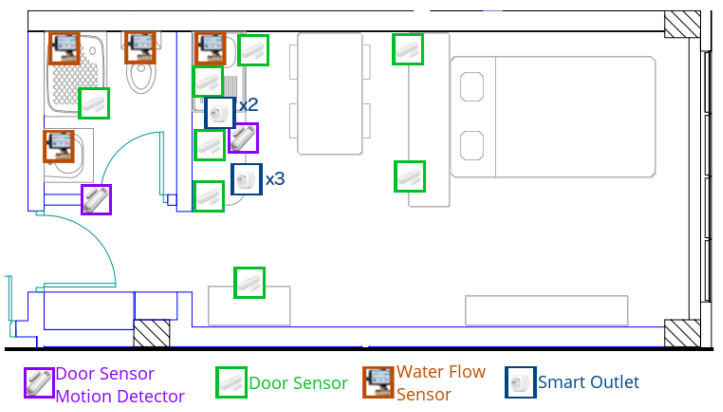
Blueprint of the flat with all the sensor positions.

**Figure 4 sensors-22-02439-f004:**
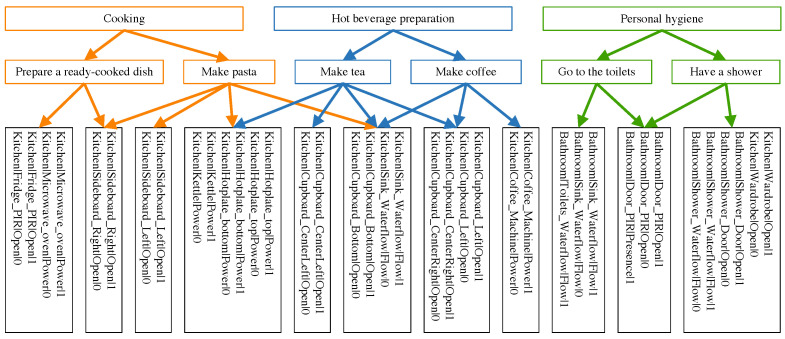
Activities, actions and events decomposition used in the living-lab.

**Figure 5 sensors-22-02439-f005:**
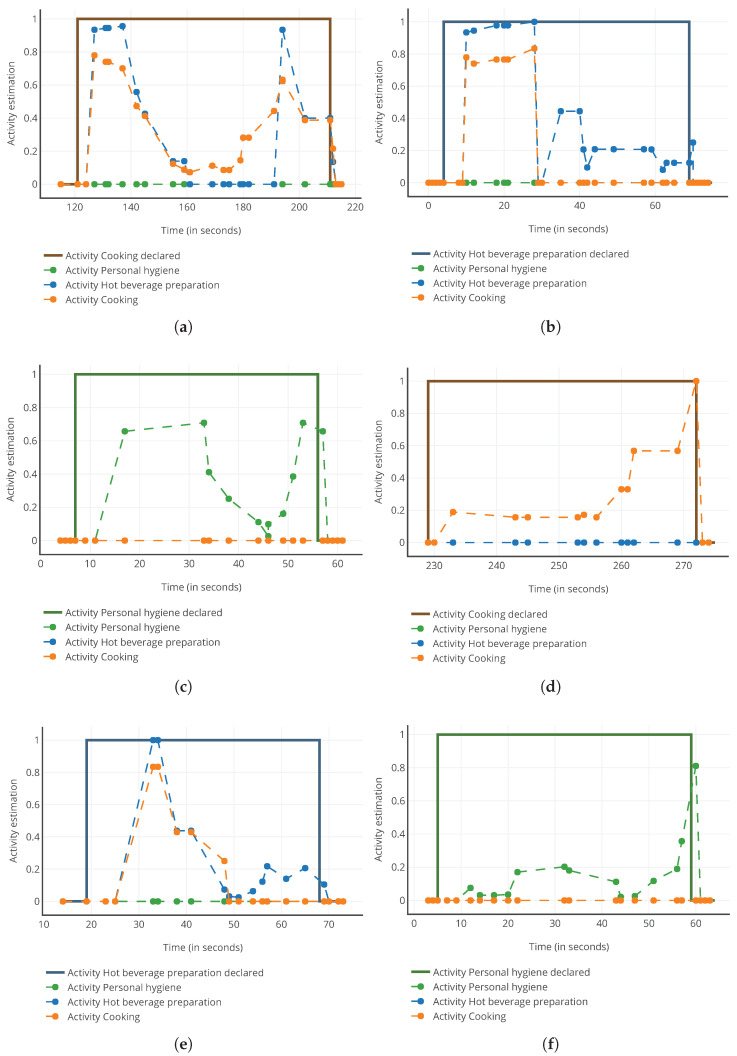
Test cases of activity recognition. (**a**) “Make pasta” test case; (**b**) “Make coffee” test case; (**c**) “Go to the toilets” test case; (**d**) “Prepare a ready-cooked dish” test case; (**e**) “Make tea” test case; (**f**) “Have a shower” test case.

**Table 1 sensors-22-02439-t001:** Point of view and targeted level of AAE decomposition by each metric.

Metric	Point of View			Targeted Level		
	Activity	Action	Event	Activity	Action	Event
$C_{a}\left(e_{k}|A_{i}\right)$	-	-	✓	✓	✓	-
$C_{a}\left(e_{k}\right)$	-	-	✓	-	✓	-
$C_{A}\left(e_{k}\right)$	-	-	✓	✓	-	-
$C_{A}\left(a_{j}\right)$	-	✓	-	✓	-	-
$S_{e}\left(a_{j}|a_{k}\right)$	-	✓	-	-	-	✓
$S_{a}\left(A_{i}|A_{j}\right)$	✓	-	-	-	✓	-
$S_{e}\left(A_{i}|A_{j}\right)$	✓	-	-	-	-	✓
$W(e_{k}|a_{j})$	-	-	✓	-	✓	-
$W(a_{j}|A_{i})$	-	✓	-	✓	-	-
$W(e_{k}|A_{i})$	-	-	✓	✓	-	-
$F(e_{k}|A_{i})$	-	-	✓	✓	-	-
$F(a_{j}|A_{i})$	-	✓	-	✓	-	-
$D_{e}\left(A_{i}\right)$	✓	-	-	-	-	✓
$D_{a}\left(A_{i}\right)$	✓	-	-	-	✓	-

**Table 2 sensors-22-02439-t002:** Evaluation of activity-related metrics for each activity in the case study.

	$S_{a}\left(A_{i}|A_{j}\right)$			$S_{e}\left(A_{i}|A_{j}\right)$			$D_{a}\left(A_{i}\right)$	$D_{e}\left(A_{i}\right)$
	$A_{C}$	$A_{\mathrm{HBP}}$	$A_{\mathrm{PH}}$	$A_{C}$	$A_{\mathrm{HBP}}$	$A_{\mathrm{PH}}$		
** $A_{C}$ **	1	0.0	0.0	1	0.56	0.0	1.0	0.69
** $A_{HBP}$ **	0.0	1	0.0	0.56	1	0.0	1.0	0.76
** $A_{TCPH}$ **	0.0	0.0	1	0.0	0.0	1	1.0	1.0

**Table 3 sensors-22-02439-t003:** Evaluation of event-related metrics for each event in the case study.

Event	$C_{a}\left(e_{k}|A_{i}\right)$			$C_{A}\left(e_{k}\right)$	$W(e_{k}|A_{i})$			$F(e_{k}|A_{i})$		
	$A_{C}$	$A_{\mathrm{HBP}}$	$A_{\mathrm{PH}}$		$A_{C}$	$A_{\mathrm{HBP}}$	$A_{\mathrm{PH}}$	$A_{C}$	$A_{\mathrm{HBP}}$	$A_{\mathrm{PH}}$
Bathroom|Door_PIR|Open|0	0	0	2	1	0	0	0.14	0	0	1.0
Bathroom|Door_PIR|Open|1	0	0	2	1	0	0	0.14	0	0	1.0
Bathroom|Door_PIR|Presence|1	0	0	2	1	0	0	0.14	0	0	1.0
Bathroom|Sink_waterflow|Flow|1	0	0	1	1	0	0	0.08	0	0	1.0
Bathroom|Sink_waterflow|Flow|1	0	0	1	1	0	0	0.08	0	0	1.0
Bathroom|Shower_waterflow|Flow|1	0	0	1	1	0	0	0.06	0	0	1.0
Bathroom|Shower_waterflow|Flow|0	0	0	1	1	0	0	0.06	0	0	1.0
Bathroom|Toilets_waterflow|Flow|1	0	0	1	1	0	0	0.08	0	0	1.0
Bathroom|Shower_Door|Open|0	0	0	1	1	0	0	0.06	0	0	1.0
Bathroom|Shower_Door|Open|1	0	0	1	1	0	0	0.06	0	0	1.0
Kitchen|Cupboard_Bottom|Open|0	1	2	0	2	0.04	0.08	0	0.33	0.67	0
Kitchen|Cupboard_Bottom|Open|1	1	2	0	2	0.04	0.08	0	0.33	0.67	0
Kitchen|Cupboard_CenterLeft|Open|0	0	1	0	1	0	0.03	0	0	1.0	0
Kitchen|Cupboard_CenterLeft|Open|1	0	1	0	1	0	0.03	0	0	1.0	0
Kitchen|Cupboard_CenterRight|Open|0	0	2	0	1	0	0.08	0	0	1.0	0
Kitchen|Cupboard_CenterRight|Open|1	0	2	0	1	0	0.08	0	0	1.0	0
Kitchen|Cupboard_Left|Open|0	0	2	0	1	0	0.08	0	0	1.0	0
Kitchen|Cupboard_Left|Open|1	0	2	0	1	0	0.08	0	0	1.0	0
Kitchen|Fridge_PIR|Open|0	1	0	0	1	0.08	0	0	1.0	0	0
Kitchen|Fridge_PIR|Open|1	1	0	0	1	0.08	0	0	1.0	0	0
Kitchen|Kettle|Power|1	1	1	0	2	0.04	0.03	0	0.5	0.5	0
Kitchen|Kettle|Power|0	1	1	0	2	0.04	0.03	0	0.5	0.5	0
Kitchen|Coffee_machine|Power|1	0	1	0	1	0	0.05	0	0	1.0	0
Kitchen|Coffee_machine|Power|1	0	1	0	1	0	0.05	0	0	1.0	0
Kitchen|Microwave_oven|Power|1	1	0	0	1	0.08	0	0	1.0	0	0
Kitchen|Microwave_oven|Power|0	1	0	0	1	0.08	0	0	1.0	0	0
Kitchen|Hotplate_top|Power|1	1	1	0	2	0.04	0.03	0	0.5	0.5	0
Kitchen|Hotplate_top|Power|0	1	1	0	2	0.04	0.03	0	0.5	0.5	0
Kitchen|Sink_waterflow|Flow|1	1	2	0	2	0.04	0.08	0	0.33	0.67	0
Kitchen|Sink_waterflow|Flow|0	1	2	0	2	0.04	0.08	0	0.33	0.67	0
Kitchen|Hotplate_bottom|Power|1	1	1	0	2	0.04	0.03	0	0.5	0.5	0
Kitchen|Hotplate_bottom|Power|0	1	1	0	2	0.04	0.03	0	0.5	0.5	0
Kitchen|Sideboard_Left|Open|0	1	0	0	1	0.04	0	0	1.0	0	0
Kitchen|Sideboard_Left|Open|1	1	0	0	1	0.04	0	0	1.0	0	0
Kitchen|Sideboard_Right|Open|0	2	0	0	1	0.12	0	0	1.0	0	0
Kitchen|Sideboard_Right|Open|1	2	0	0	1	0.12	0	0	1.0	0	0
Kitchen|Wardrobe|Open|0	0	0	1	1	0	0	0.06	0	0	1.0
Kitchen|Wardrobe|Open|1	0	0	1	1	0	0	0.06	0	0	1.0

## Data Availability

Not applicable.

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
