# Peer review of "Mathematical Criteria for a Priori Performance Estimation of Activities of Daily Living Recognition"

_sensors, 2022, doi:10.3390/s22072439_

Round 1

Reviewer 1 Report

The authors need to take in consideration the following suggestions before to accept it:

(1)       The introduction needs to be improved discussing current articles because only a few new articles have been included in it.

(2)       The contribution needs to be re-written because it is not clear.

(3)       I do not perceive a clear contribution since I observe a huge quantity of equations and remarks, which are not related clearly for active reignition. For this reviewer, this huge quantity of equations and remarks do not exhibit a clear contribution to the subject.

(4)       It is mandatory that the authors compare their results with other methodologies in order to validate or show that their contribution is important in the subject.

(5)       The conclusion needs to be improved, adding quantitative results not only qualitative results. In addition, it is important to mention, what is the next with the investigation?

Author Response

Dear reviewer,

Thank you very much for your evaluation.

We made some modifications (in blue) to take into account your comments.

In particular, we add a paragraph at the end of the introduction to resume the contribution.

Unfortunately, it is not possible to compare our proposition with other methodologies. Indeed, AAL decomposition is used in order to provide a framework to recognize activities. Many works use Hidden Markov Model, sometimes statistical models, and we use in our previous work Probabilistic Automaton.

However, in all work, the effectiveness of the contribution depends on the instrumentation, and this point is never mathematically discussed in publications.

Even the best methodologies to recognize activity could give poor results if the instrumentation is not adapted.

The goal of this work is to provide mathematical criteria to evaluate the effectiveness of the recognition process. To the best of our knowledge, the only works that proposed metrics are focused on the evaluation of the deviation detection.

As the next investigation, as it is written, future work will be devoted to scaling up the approach we proposed and applied in our analysis of the case study in order to automate it. We add a sentence to precise more effectively the current work.

Best regards,

Reviewer 2 Report

The issue of monitoring Activities of Daily Living (ADL) is one of hot research topics in the field of Machine Learning. This paper proposed mathematical criteria to evaluate a priori the performance of the instrumentations for the goals of ADL recognition. A common decomposition in order to design ADL by mathematical model, is the Activities-Actions-Events (AAE).

The contribution of this paper is fair. This paper is very well written with detailed theoretical models, and the experiment results are convincing. In this paper, metrics for an a priori estimation of the performance of model-based Activity Recognition were introduced: participation counter, sharing rate, weight, elementary contribution and distinguishability. At all the levels of the activity-action-event decomposition, they allow to determine precisely how the resulting models will behave, with their strengths and weaknesses.

However, there are also some comments which might help the author to further improve this paper.

First, in “Abstract”, the line of 6, the word “mathemtical” may be “mathematical”.

Second, in “2. Description of activities”, the line of 55, the word “requires” may be “require”.

Third, in “3.4. Sharing rates”, the line of 115, the word “activities” may be “activity”.

Fourth, in “3.4. Sharing rates”, the line of 133 and 135, the word “consider” may be “considers”.

Fifth, in the line of 169 and 179, the word “their” may be “there”.

Sixth, in “5.1. Smart flat description”, the line of 245, the word “represents” may be “represent”.

Seventh, in “5.2. Activities to monitor and proposed decomposition”, the line of 254, the word “outlet” may be “outlets”.

Eighth, the testing samples set size should be large enough to show your results are indeed convincing, reliable, and meaningful.

Ninth, there are few comparative experimental algorithms and not enough comparisons with other methods to show the effectiveness of the experiment in this paper.

Tenth, more recent and relevant publications need to be updated and added to the reference list, for examples 2019, 2020, 2021.

Generally, there is something offered in the paper, but the author should prepare a major revision for the paper.

Author Response

Dear reviewer,

Thank you very much for your evaluation helping us to improve our paper.

We made modifications (in blue) to the paper to take into account your comments.

Furthermore, we update the bibliography and add interesting recent publications.

Best regards,

Reviewer 3 Report

The paper proposes a evaluation framework for the decomposition/modeling of activities of daily living, based on the sensors available for a specific application. The approach is general enough that it could be used for any type of activity, not just in the field of AAL.

The paper is structured logically and develops the evaluation metrics in a sraight forward manner. The metrics seem very helpful and are evaluated using a small test setup. A larger setup and some automatically derived activity models would be interesting, but are not necessary to complete the paper.

Over all a very nice work and interesting contribution

Author Response

Dear Reviewer,

Thank you very much for your positive comments.

As you noticed, this approach is not only dedicated to AAL, and we currently use our work to Human-Cyber-Physical System for industry 5.0.

Best regards,

Reviewer 4 Report

The authors proposed mathematical criteria for performance estimation of recognition of activities of daily living. The approach is well explained. The normal activities are recognized. However, the authors should explain whether they also considered detection of abnormal events. Recognition of regular activities of daily living is very good. Nevertheless, different order of individual actions should be considered because it does not necessarily mean a problem. For example, in meal preparation some actions can be done in different order. But on  the other side, frequent random opening of cupboard or fridge door might indicate developing a neurodegenerative disorder. Please explain this issue.

Author Response

Dear reviewer,

Thank you very much for your evaluation.

We made some modifications (in blue) to the paper to take into account your comments.

We agree with you about the importance of the process of detecting abnormal behavior.

Indeed, in this paper, we consider the discovery, recognition, and detection of abnormal behavior are done correctly.

Then, the only parameter that changes the effectiveness of the detection is the AAL decomposition.

The hypothesis the detection is done came from our other works.

In [1]: we proposed a framework to discover and recognize activities.

In this work, we use an "envelope model" that includes all manners to perform an activity.

It is then the goal of the abnormal detector to compute if the realization of the activity is good or not. 

The detection of deviation is very recent work and is still under process, but we have already started communicating on this. In this work [2], we show we have good detection of deviations in the condition we have a good recognition of activities, whatever the manners to do them.

We propose a set of criteria presented in this paper to ensure good recognition. 

To resume, we decompose the global problem into many smalls ones : 

  1. how to optimize the instrumentation in order to discover and recognize activities
  2. how to discover and to recognize activities
  3. how detect deviations when an activity is perform.

In the beginning, we consider the first problem as solved (hypothesis), and we proposed a framework to problem 2. Then we propose criteria to solve problem 1.

For now, the problem is not completely solved, but the set of criteria ensures the good effectiveness of the recognition process. We are currently trying to propose an optimization problem, with an optimal solution, of it.

[1] K. Viard, M. P. Fanti, G. Faraut and J. -J. Lesage, "Human Activity Discovery and Recognition Using Probabilistic Finite-State Automata," in IEEE Transactions on Automation Science and Engineering, vol. 17, no. 4, pp. 2085-2096, Oct. 2020, doi: 10.1109/TASE.2020.2989226.

[2] K. Fouquet, G. Faraut and J. -J. Lesage, "Model-Based Approach for Anomaly Detection in Smart Home Inhabitant Daily Life," 2021 American Control Conference (ACC), 2021, pp. 3596-3601, doi: 10.23919/ACC50511.2021.9483053.

Round 2

Reviewer 1 Report

The authors have improved the quality of their work according to the reviewer's suggestions; however, I consider that it is very important to compare their proposal (qualitatively or quantitatively) with other recent proposals introduced in the literature in order to show the advantages and disadvantages of their proposal. 

Author Response

Dear reviewer,

Thank you very much for your evaluation.

In this work, we do not aim at assessing the performances of any activity recognition algorithm, something for which indeed a qualitative analysis against other algorithms (and even better a quantitative one) would be more than necessary.

However, we aim here at providing means to evaluate somehow the relevance of instrumentation and of AAE decomposition(s) determined based on it. (clearly stated on page 2 ll. 48-49 of the manuscript).

Regarding our proposal i.e., a framework of equations for analysis of AAE decomposition and aiming at inferring a priori performance of Activity Recognition algorithms, we have not found in the literature any close contribution.

If there is any recent proposal close to our contribution that we missed, we would be glad to have the references so that we could at least perform a qualitative analysis against our framework.

In addition, the case study presented showed that, for a considered activity recognition algorithm, the framework of equations is relevant to determine a priori the results likely to be obtained with the AR algorithm. It could be possible to consider other AR algorithms or to reconsider the performances of the chosen AR algorithm based on a different AAE decomposition. We consider this would not significantly increase the value of the contribution as in the first case this would scramble the objective of our paper which is not performance evaluation of AR algorithms, and in the second case, this would be very case-specific.

Therefore, we chose not to perform any additional modifications to our manuscript.

Best regards,

Reviewer 2 Report

All questions have been corrected. It is a qualified paper. Agree to publish.

Author Response

Dear Reviewer,

Thank you again for your comments helping us improve our paper.

Best regards,